# An Innovative Method for BTEX Emission Inventory and Development of Mitigation Measures in Developing Countries—A Case Study: Ho Chi Minh City, Vietnam

**DOI:** 10.3390/ijerph192316156

**Published:** 2022-12-02

**Authors:** Quoc Bang Ho, Hoang Ngoc Khue Vu, Thoai Tam Nguyen, Thi Thao Nguyen Huynh

**Affiliations:** 1Institute for Environment and Resources (IER), 142 To Hien Thanh St., Dist. 10, HCMC, Ho Chi Minh City 700000, Vietnam; 2Department of Academic Affairs, Vietnam National University, Ho Chi Minh City 700000, Vietnam

**Keywords:** air quality modeling, BTEX, emission inventory, Ho Chi Minh City, low- and middle-income countries (LMICs)

## Abstract

Benzene, toluene, ethylbenzene, and xylenes (BTEX) are carcinogenic pollutants. However, the average concentration in 1 h of some pollutants belonging to BTEX, such as benzene, in Ho Chi Minh City (HCMC) is higher than the national standard QCVN 06:2009/BTNMT by about ten times. This research is the first to calculate the emission of BTEX for developing countries on a city scale. This paper developed a method to calculate cold emission factors based on hot emission factors for BTEX. Five spreadsheets developed and calculated these cold emission factors for five vehicle categories. A comprehensive emission inventory (EI) for BTEX was conducted in HCMC to determine the cause of BTEX pollution. An innovative methodology with bottom-up and top-down combination was applied to conduct BTEX EI, in which the EMISENS model was utilized to generate the EI for road traffic sources, and the emission factors method was utilized for other emission sources. Among emission reasons, motorcycles contribute the highest to HCMC air pollution, responsible for 93%, 90%, 98.9%, and 91.5% of benzene, toluene, ethylbenzene, and xylene, respectively. Cars contributed 5%, 6%, 0.8%, and 6.5% of benzene, toluene, ethylbenzene, and xylene, respectively. For LDVs, the emission from benzene, toluene, ethylbenzene, and xylene accounted for 1%, 2%, 0.2%, and 1.9%. The major reasons for point sources were metal production, which had 1%, 2%, and 0.1% for benzene, toluene, ethylbenzene, and xylenes emissions. The area source had a minority emission of total BTEX in Ho Chi Minh City. Our findings can be used to invest in the most significant sources to reduce BTEX in HCMC. Our approach can be applied in similar urban areas in BTEX EI. This research also developed nine measures to reduce BTEX in HCMC for human health protection.

## 1. Introduction

Since the eighties, dozens of programs have been carried out to study and detect benzene, toluene, ethylbenzene, and xylenes (BTEX) in atmospheric air. Currently, hazardous VOC emissions are routinely controlled by a network of monitoring stations located in areas with high exposure to these hazardous substances [1]. Exposure to high concentrations of BTEX can cause problems with the skin and senses, depression, and impacts on the respiratory system [2,3,4]. In the group of BTEX, benzenes get special attention because they can cause cancer. BTEX originates from many man-made sources, such as fossil fuels, motor vehicles, and industrial production related to petrochemicals, paints, solvents, synthetic resins, and synthetic rubbers [5]. In addition to harmful effects on health, BTEX are also substances that affect the environment. BTEX are volatile compounds, so they are easily dispersed in the air. At permissible concentrations, BTEX are not harmful to the environment, but high concentrations will cause significant impacts on the environment. BTEX will harm the ecosystem if they enter the environment due to breakage or leakage from containers. BTEX in the air will react with many other pollutants, increasing their toxicity to the environment. In particular, BTEX are involved in the formation of ozone (which is a strong oxidant that creates many other pollutants), increase the ozone content in the air, and participate in photochemical reactions to form photochemical smog [6].

The studies were carried out in different types of environments such as rural and urban, bus, tunnel, indoor, etc. Some studies on BTEX concentration in an urban environment can be mentioned, such as the study by [7,8,9,10]. Studies on human exposure to BTEX have also attracted the attention of the scientific community with a wide variety of subjects, from gas station employees, street vendors, office workers, bus drivers, etc. [11,12]. Fuel combustion activities are the main cause of these pollutants.

In Vietnam, air pollution is one of the severe issues, especially in megacities such as Hanoi and Ho Chi Minh City (HCMC) with PM and VOCs [13]. However, BTEX pollution has been less considered. These previous studies mainly determined the concentration of BTEX at a few small roadside locations or in households and parking lots [14,15]. Preliminary studies show that benzene concentrations almost exceed Vietnamese standards. Traffic activity, especially motorbikes, is the primary source of BTEX generation in Hanoi [16] and Ho Chi Minh City [17]. However, the pollution sources from point sources (i.e., industry) and area sources (i.e., domestic activities) have been neglected.

Several studies highlighted the human health effects regarding exposure to BTEX in low- and middle-income countries (LMICs). Nevertheless, few studies have been conducted to comprehensively quantify inventory and determine the contribution of these emissions to BTEX concentrations in the air. The study aims to quantify the concentrations of BTEX from traffic sources, point sources, and area sources. We determined the contribution of emission sources to BTEX pollution in the air by the emission inventory approach. We chose Ho Chi Minh City (HCMC) in Vietnam as our case study.

HCMC is the most populous city in Vietnam with around 10 million people. There has been a rapid increase in private vehicles, especially motorcycles, since they are affordable and convenient for residents. Additionally, HCMC has the important role of economic center, with more than 19 industrial zones, 30 industrial clusters, and many factories and enterprises. These activities cause high pressure on air pollution. Additionally, domestic cooking, other transportation activities such as airports, railways, etc., also contribute to air pollutant emissions. However, these contamination sources have rarely been considered in terms of identifying main pollution sources. In this study, we used bottom-up and top-down approaches to analyze the BTEX inventory from these sources for HCMC.

## 2. Materials and Methods

Sources of emissions in HCMC are divided into three major groups: point sources, area sources, and line sources. One of the main contributions of air pollution is point source with stationary chimneys. Area sources are distributed evenly in space and produce fewer emissions. Traffic activities on roads, railways, airports, and ports are line sources. In this research, the emission factor methodology was applied to calculate air emissions for area sources and point sources. We used EMISENS model to calculate on-road mobile source. Additionally, we applied the SPD—GIZ model to analyze the emission from ports.

### 2.1. Line Source

#### 2.1.1. On-Road Emission Sources

For urban areas, road traffic contributes to significant BTEX emissions. However, road traffic is the most challenging source for calculating emissions. Therefore, in this research, we applied an innovative method by combining bottom-up and top-down approaches to calculate total BTEX emissions. Cold (Ecold), hot (Ehot), and evaporation (Eevap) emissions are three stages that we considered. The stages showed three periods of engines, including warming up, stable reaching heat, and releasing VOCs into the air. Cold emission accounts for about 20–30% of the total emission of pollutants. Many previous projects about emission inventory estimate emissions by only top-down approaches. For some unavailable data of emission factors, we applied the COPERT IV theory and developed the method to calculate the cold emission factor for road traffic. We also included this approach for BTEX.

##### Calculation of the Averaged Cold Emission Factors

Measurement campaigns are applied to estimate the averaged hot emission factors per vehicle category, street, and pollutants. We calculated the average emission factors for a certain number of vehicle categories (e.g., car, light truck, heavy truck, bus, and motorcycle) in Equations 1, 2 and 3 using the associated fleet composition. The fleet composition is determined by using the results of the survey.

Hot emission factors are calculated as follows:(1)$$e_{Iv,is}^{hot}=\sum_{iv}^{nIv}\alpha_{iv}e_{iv,is}^{hot}$$
where $\alpha_{iv}$ is the proportion of each type of vehicle in each category (Equation (1)).

Cold emission factors are illustrated as follows:(2)$$e_{iv,is}^{cold}=e_{iv,is}^{hot}\left(\frac{e_{iv,is}^{cold}}{e_{iv,is}^{hot}}-1\right)\;\mathrm{with}\;\frac{e_{iv,is}^{cold}}{e_{iv,is}^{hot}}=A_{iv}V+B_{iv}T+C_{iv}$$
where the *A*, *B,* and *C* (Equation (2)) constants depend on the vehicle type, *V* and *T* are the cold speed (km/h) and atmospheric temperature (°C).
(3)$$e_{Iv,is}^{cold}=\sum_{iv}^{nIv}e_{iv,is}^{hot}\left(\frac{e_{iv,is}^{cold}}{e_{iv,is}^{hot}}-1\right)=\sum_{iv}^{nIv}e_{iv,is}^{hot}\left(A_{iv}V+B_{iv}T+C-1\right)$$
(4)$${\bar{e}}_{Iv,is}^{cold}={\left(\bar{e^{hot}A}\right)}_{Iv,is}V_{is}+{\left(\bar{e^{hot}B}\right)}_{Iv,is}T+{\left(\bar{e^{hot}C_{1}}\right)}_{Iv,is}$$
where ${\bar{e}}_{Iv,is}^{cold}$ is the average cold emission factor for the vehicle category Iv, and nIv is the number of vehicles in the category Iv (Equation (4)).

##### Calculation of the BTEX Emissions

The emissions are computed on each cell per hour:(5)$$E^{hot}\left(x,y,t\right)={\bar{e}}_{Iv,Is}^{hot}{\left[FL\right]}_{Iv,Is}\left(x,y,t\right)$$
(6)$$E^{cold}\left(x,y,t\right)=\left(\bar{e_{Iv,Is}^{hot}A}V_{Is}+\bar{e_{Iv,Is}^{hot}B}T+\bar{e_{Iv,Is}^{hot}C_{1}}\right){\left[\beta FL\right]}_{Iv,Is}\left(x,y,t\right)$$

These theories and equations were developed by 5 spread sheets for 5 vehicle categories. Then, we added the value in Equations 5 and 6 to the EMISENS input file [18]. The input data of EMISENS, including the collected data and emission factors, are described as follows. The collected data included road data with street length and type, the flow and conditions of traffic, diurnal traffic curves, average speed, and vehicle data. Additionally, the vehicle data were type and technology of vehicle, vehicle share, daily mileage, and fuel use. We applied the EMISENS model developed by a combination of bottom-up and top-down approaches for traffic activities (on-road mobile sources).

##### Traffic Counts

We classified streets with highways, rural roads, urban streets, suburban streets, and industrial streets (located in the industrial zone). Vehicles included motorcycles, cars, buses/coaches, heavy-duty vehicles (HDVs), and light-duty vehicles (LDVs) which weigh more and less than 3.5 tons, respectively.

Manual counting methods were conducted for traffic counting. Firstly, we measured from 6 am to 7 pm, 15 and 30 min each hour for traffic counting at sites. Secondly, we extrapolated into one hour for the traffic flows. Additionally, we used a 24 h camera for one street and counted the traffic flows based on this video.

A total of 70 streets in Ho Chi Minh City were surveyed in all five categories of the street. Counting vehicle traffic included locations with frequent traffic jams and main traffic locations in the city. Vehicle counts were on Monday for the week and two weekends (Saturday and Sunday) to obtain traffic movement for 24 h and weekdays and weekends. Motorcycles were prevalent as 80% of vehicles. Cars and other vehicles accounted for 10% for each type, respectively. While heavy traffic was from 6 am to 6 pm, HDV had peaks at night.

##### Survey of the Vehicles (On-Road)

The survey was conducted to collect the data for the characterization of five vehicles, including vehicle category, vehicle age, fuel type, engine size or loading capacity, mileage of the vehicle, and the number of trips per day.

The sample size for questionnaires: The number of respondents in the survey was 1080 for a 97% confidence level as referencing the sample size determined in Yamane’s formula.

The questionnaires: There were 3694 interviews, including 2924 interviews from Ho’s study [19] and 770 new interviews. The number of interviews for each type of vehicle is in Appendix A.

##### Emission Factors (EFs)

The EFs for benzene, toluene, ethylbenzene, and xylenes in road traffic were considered in this study. The data were referenced from the EFs of India [20,21] and the EFs of other countries in CORINAIR [22,23].

#### 2.1.2. Non-Road

The non-road mobile sources included the airport and railway station, and the EI approach followed US EPA guidelines.

##### Airports

Equation to calculate emissions from the airport:E = (LTO × EF)/1000(7)
where E is emissions (tons/year), LTO is the number of each type of airplane in one trip/year, and EF is emission factors relating to LTO (kg pollutant/trip).

We collected flight data from HCMC’s Department of Transportation and the schedules of airlines operating in the city. The information is shown in Appendix A. The sources of EFs were taken from the NPI (2001) Emission Estimation Technique Manual for Airports [24].

##### Railway Stations

Equation to calculate emissions from stations:E = (A × EF)/1,000,000(8)
where E is emissions (tons/year), A is the amount of fuel use (kg/year), and EF is emission factors relating to A (g pollutant/kg fuel).

We collected the data of trips from Hoa Hung railway station. In addition, we interviewed six railway drivers to obtain information about the amount of fuel consumed at the stations. EFs data were referenced from the Emissions Estimation Technique Manual for Aggregated Emissions from Railways [25].

### 2.2. Area Source

We conducted surveys at 24 districts to collect data about the amount of fuel consumption and fuel type. There were 2593 questionnaires completed at 2,061,959 households and 261 restaurants to collect the activity data of generated emissions. There are around 5069 restaurants in Ho Chi Minh City.

For emissions from area sources, the general formula is based on emission factors and activities (Equation (9)).
E = A × EF × [1 − (ER/100)](9)
where E is emissions (normally in tons/year), A is activity rate (amount of fuel use, capacity, or number of products), EF is emission factors (related to A), and ER is emission reduction efficiency (only if abatement devices are used). The sources of EFs were taken from the National Pollutant Inventory (NPI) in Australia. BTEX of area sources emits from burning LPG and charcoal briquettes for cooking.

### 2.3. Point Sources

The general formula was applied to calculate emissions from point sources which were based on emission factors and activities. Data of emissions including fuel types, amount of fuel use, output products, technology, air pollution treatment, etc., were collected from the Department of Natural Resources and Environment and People’s Committees of Districts at HCMC. There were 753 questionnaires conducted for all districts at 753 factories representing all industrial sectors in HCMC. The rest of the 2053 factories we collected from the General Statistics Office. Then, emissions were calculated based on related EFs referenced from the National Pollutant Inventory (NPI) in Australia [26,27,28,29] and EFs in China [30]. The BTEX emissions are calculated as the sum of the two results:Emissions by sector: calculated by multiplying the emission factor of each industry by the amount of product produced.Emissions when burning fuel: calculated when multiplying the emission factor of each pollutant by the total amount of fuel used, then multiplying by % of the treatment efficiency of that system (if any).

## 3. Results and Discussion

### 3.1. Emissions from Traffic

#### 3.1.1. On-Road Emission Sources

Motorcycles were the majority emission source, accounting for more than 90% of the total road traffic emissions. Toluene contributed the most BTEX for traffic activities, followed by ethylbenzene, xylene, and benzene (Table 1).

#### 3.1.2. Other Emission Sources

Railway stations and airport emissions are presented in Table 2.

The results show that on-road (especially motorcycles) was the main contribution of traffic emission in HCMC, accounting for nearly 99% of all BTEX pollutants. Airports and railways had a small contribution to the BTEX emission with less than 0.1%.

### 3.2. Emissions from Industry

The emissions of each manufacturing facility were calculated from two sources, including process emissions and emissions originating from combustion activities within the industry. Textile, metal production, food, and plastic were the main pollution sectors. Coal was the largest amount of fuel used in industry. Oil types (DO and FO), gasoline, wood, and wood products were included.

Toluene and xylenes emissions are the major pollutants, followed by benzene and ethylbenzene (Table 3). The manufacturing of iron and steel (metal production), textile, food, printing, and beverages are relatively large BTEX emitters compared with other sectors.

The contribution of all industrial sectors of the point sources in HCMC for BTEX is presented in Figure 1. Most sectors have the most significant proportion of emissions from toluene. Benzene is prevalent in some sectors such as textile, printing, and fodder. Xylenes have a large contribution to BTEX for construction and paper production. Ethylbenzene had small emissions compared with the other pollutants.

Fractions of different districts in Figure 2 show that suburban communities with industrial and manufacturing zones contributed primarily to the emission, such as Thu Duc, Tan Phu, Binh Chanh, and Hoc Mon. These districts have the majority of emissions because these areas are mainly industrial zones with relatively strong development in terms of industries. Toluene is prevalent in Binh Chanh, Tan Phu, and Thu Duc, with around 70% of total BTEX emission, while benzene and toluene had similar contributions in the Hoc Mon district with 42% of total BTEX emission (Figure 3).

Five sectors contribute the most EI to BTEX. While most BTEX sources come from producing metal (nearly 99%), the textile, food, printing, and beverage industries mainly emit BTEX through combustion. The common materials in those sectors are DO, FO, gas, and charcoal. Table 4 below shows the emission (kgs/year) for these significant sectors due to producing and burning materials.

### 3.3. Emissions from the Area Sources

The emission from households in different districts is presented in Table 5. It can be seen that BTEX emissions from households are mainly benzene and toluene, mainly in District 7, District 9, and the Binh Thanh and Go Vap Districts.

The emission from restaurants in Ho Chi Minh City is shown in Table 6 below. The restaurants had less BTEX emission compared with household activities. The total benzene, toluene, ethylbenzene, and xylenes emissions of the area source (i.e., both households and restaurants) is 20 tons/year, 4 tons/year, and 1 ton/year (same for ethylbenzene and xylenes), respectively. The benzene emission is dominant out of the four pollutants in area sources with 78%.

### 3.4. Total Emissions

#### Total Emission Results

The line source has the highest contribution of BTEX with up to 98% (Table 7). Industrial activities contributed 0.7%, 2.2%, 0.1%, and 0.8% of the total benzene, toluene, ethylbenzene, and xylenes emissions of HCMC, respectively. The area sources accounted for less than 0.4% of the BTEX. The non-road traffic contributes little to BTEX emission, with less than 0.1% BTEX.

The contribution of each BTEX pollutant is different among the activities (Figure 4). The contributions of benzene, toluene, ethylbenzene, and xylenes inline sources are similar at 19%, 31%, 27%, and 24%, respectively. While benzene has the highest contribution for BTEX in area source (78%), toluene is prevalent in point source (67%). Benzene and xylenes have similar emissions in non-road traffic, at around 40%.

We indicated the contribution of primary pollution sources in Table 8. Among emission reasons, motorcycles were the most significant contributor to HCMC air pollution, being responsible for 93%, 90%, 98.9%, and 91.5% of benzene, toluene, ethylbenzene, and xylene, respectively. Cars contributed 5%, 6%, 0.8%, and 6.5% of benzene, toluene, ethylbenzene, and xylene, respectively. For LDVs, the emissions from benzene, toluene, ethylbenzene, and xylene accounted for 1%, 2%, 0.2%, and 1.9%. The main reasons for point sources were metal production, which had 1%, 2%, and 0.1% for benzene, toluene, ethylbenzene, and xylenes emissions. The area source had a minority emission of BTEX.

The findings of this study about the main source of BTEX were similar to other studies. For example, Rad et al. (2014) indicated transportation and industry were the primary pollution sources of BTEX in the metropolitan city of Iran [31]. Cui et al. (2012) highlighted the fuel use and diesel exhaust sources of air pollution in Beijing, China [32]. In Ho Chi Minh City (Vietnam), motorcycles contributed the most to the traffic fleet [33]. The motorcycle exhaust origins of BTEX were indicated in the study of Tran and Pham (2012). Therefore, the results in our study are appropriate given the relevant research. They emphasize the importance of controlling these main pollution sources. The next section provides some mitigation measures that can be feasible to invest in HCMC.

### 3.5. Mitigation Measures

In this section, we developed measures for BTEX reduction based on the results of the detailed BTEX emission inventory in Table 8:Air smoke checking automobiles randomly on the road: HCMC has checked for a few days for buses. As a result, 22% of the buses were checked for air smoke exceeding the acceptable standard. It is necessary to carry out random and continuous exercises with all cars.Air smoke checking for motorcycles: HCMC has measured several motorcycles, as a result, the emission from motorcycles will be reduced by 30%.Develop a project for public transportation: To conduct this measure, there are several steps that need to be carried out: reviewing and assessing the short-term financial support required to establish new bus teams and routes; a government agency will draft the necessary regulations to facilitate the financial support for new routes; new rules approved by the HCMC People’s Committee; implementation and inspection; and developing a plan to expand the bus network.Develop a bike sharing system.Inspect outdated motorcycles and remove them: Investigate and review statistics on the number of motorcycles, three-wheeled motorcycles, and four-wheelers transporting passengers and goods; develop regulations to suspend the vehicles that have not warranted technical safety.Replace cleaners and cook stoves for households.Publicly raise awareness of air pollution.Automatic air quality monitoring system: At least 15 stations for Ho Chi Minh City and develop a warming system for human health protection such as the Healthy Air system.Study air emission receiving zones: This measure can produce a map of the air pollution reception area for HCMC. This map serves as a basis for the city’s socio-economic development planning.

## 4. Conclusions

This paper developed a method with top-down and bottom-up approaches to calculate cold emission factors based on hot emission factors for BTEX. Five spreadsheets developed and calculated these cold emission factors for five vehicle categories separately. A total of 4383 questionnaires were carried out in this study (including 770, 6, 2593, 261, and 753 questionnaires for on-road, railway drivers, residents, restaurants, and factories, respectively) to estimate emissions, including line, point, area, and biogenic sources. Additionally, we counted the traffic activities of 70 streets. The results show that vehicle exhausts were the main BTEX emission source, accounting for 98–99%. Motorcycles were the most significant contributors to HCMC air pollution, being responsible for 93%, 90%, 98.9%, and 91.5% of benzene, toluene, ethylbenzene, and xylene, respectively. Cars contributed 5%, 6%, 0.8%, and 6.5% of benzene, toluene, ethylbenzene, and xylene, respectively. The point source has the most contribution from metal production with nearly 99% BTEX emission compared with other sectors. Due to industrial emissions, Thu Duc, Binh Chanh, Tan Phu, and Hoc Mon have the most BTEX pollution in Ho Chi Minh City. The area sources contributed less BTEX emissions. Most of the emission originating from households are derived from fossil fuel burning. Therefore, Ho Chi Minh City needs to consider these results when developing a strategy for air pollution reduction. The research also developed nine measures to reduce BTEX in HCMC for human health protection.

## Figures and Tables

**Figure 1 ijerph-19-16156-f001:**
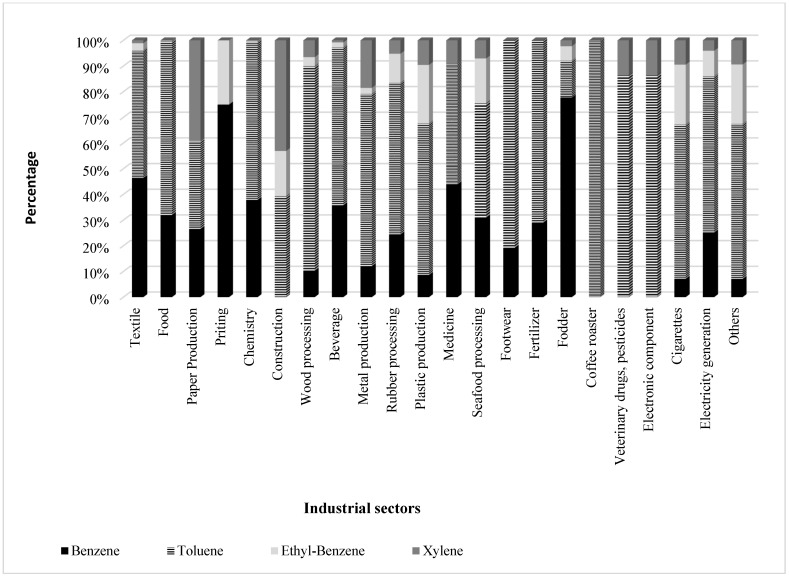
Contribution of BTEX emission load (kg/year) of industrial sectors of the point sources due to combustion fuel use and production by industry in HCMC.

**Figure 2 ijerph-19-16156-f002:**
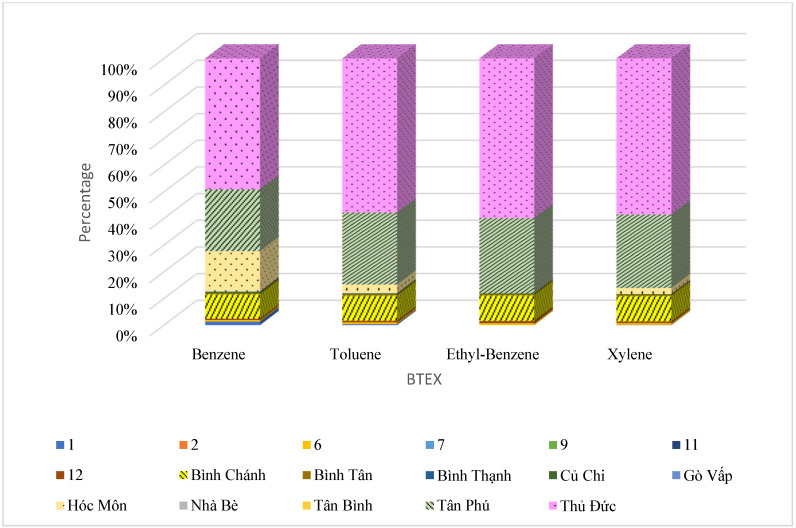
Contribution of BTEX emission source from districts in Ho Chi Minh City.

**Figure 3 ijerph-19-16156-f003:**
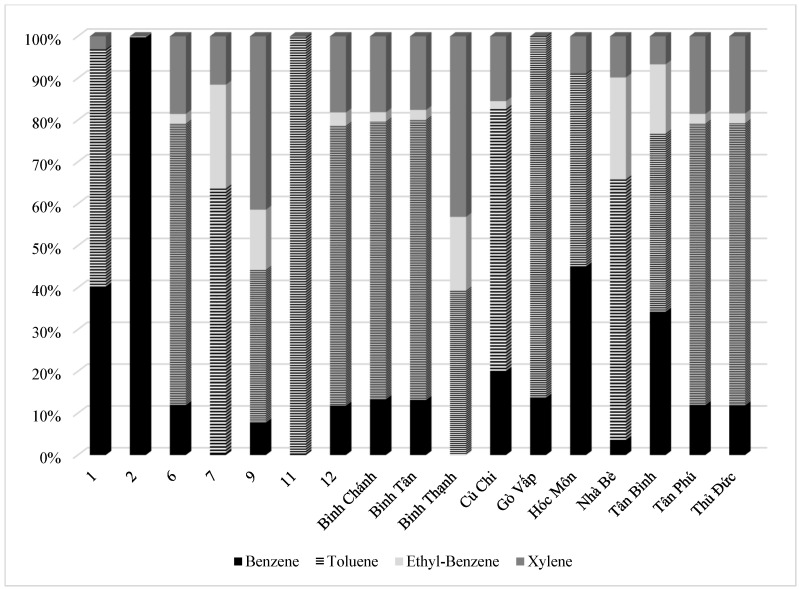
Contribution of BTEX emission load due to combustion fuel use and production of industry in each district in Ho Chi Minh City.

**Figure 4 ijerph-19-16156-f004:**
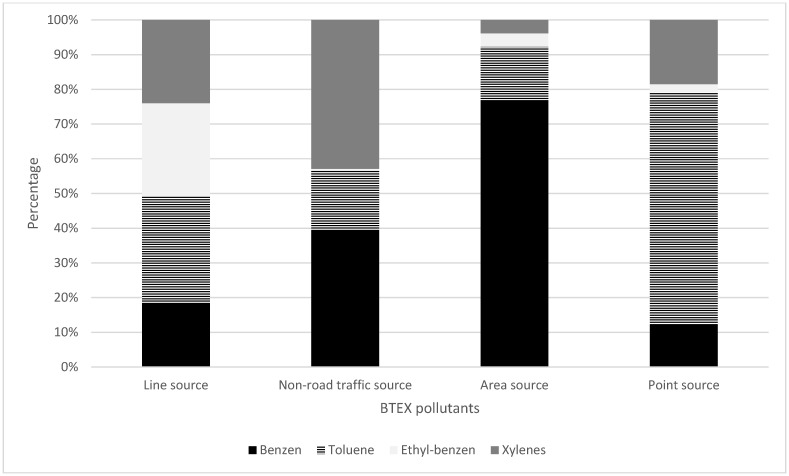
Contribution of each BTEX pollutant to the EI of different sources in HCMC.

**Table 1 ijerph-19-16156-t001:** Emission from on-road sources for five transportation means (tons/year).

Vehicles	Pollutants (Tons/Year)			
	Benzene	Toluene	Ethylbenzene	Xylene
Buses	5.16	11.32	1.66	10.09
Motorcycles	4090.61	6660.35	6253.91	5191.92
Cars	223.08	469.53	49.93	368.61
LDVs	64.82	136.44	14.51	107.11
HDVs	3.58	7.86	1.15	7.01
Total	4387.25	7285.49	6321.15	5684.74

**Table 2 ijerph-19-16156-t002:** Emission From Traffic.

Emission Sources	Pollutants (Tons/Year)			
	Benzene	Toluene	Ethylbenzene	Xylenes
On-road	4387.25	7285.49	6321.15	5684.74
Airport	2.53	1.11	-	2.78
Railway	0.027	0.001	0.028	0.004
Total	4389.8	7286.6	6321.2	5687.5

**Table 3 ijerph-19-16156-t003:** BTEX emission (kgs/year) from industrial sources due to producing and combustion activities in Ho Chi Minh City in 2021.

No.	Sectors	Emission (kgs/Year)			
		Benzene	Toluene	Ethylbenzene	Xylene
1	Textile	58.04	62.22	3.51	1.42
2	Food	57.58	121.97	0.58	0.24
3	Paper Production	24.16	31.47	0.11	35.75
4	Printing	45.95	0.05	15.32	0.00003
5	Chemistry	26.22	42.74	0.29	0.12
6	Construction	0.01	5.10	2.29	5.60
7	Wood processing	0.001	0.01	0.001	0.001
8	Beverage	46.28	79.67	2.58	1.04
9	Metal production	29,144.74	162,656.92	5773.61	44,943.96
10	Rubber processing	5.02	12.15	2.34	1.08
11	Plastic production	0.01	0.09	0.03	0.01
12	Medicine	0.009	0.014	0.005	0.002
13	Seafood processing	0.01	0.01	0.01	0.002
14	Footwear	8.71	36.89	0.000001	0.0000005
15	Fertilizer	0.0005	0.001	0	0
16	Fodder	0.05	0.01	0.0034	0.0014
17	Coffee roaster	0	0.01	0	0.0000
18	Veterinary drugs, pesticides	0	0.00007	0	0.00001
19	Electronic component	0	0.00000001	0	0.000000002
20	Cigarettes	0.0000005	0.0000040	0.0000016	0.0000006
21	Electricity generation	4.99	12.09	1.99	0.81
22	Others	0.002	0.02	0.01	0.003
Total		29,421.77	163,061.43	5802.68	44,990.03

**Table 4 ijerph-19-16156-t004:** BTEX emission (kgs/year) from producing and combustion activities of five primary industrial sectors in Ho Chi Minh City.

Sectors	Emission (kgs/Year)							
	From Producing				From Combustion			
	Benzene	Toluene	Ethylbenzene	Xylenes	Benzene	Toluene	Ethylbenzene	Xylenes
Textile	-	0.001	-	-	58.04	62.22	3.51	1.42
Food	-	-	-	-	57.58	121.97	0.58	0.24
Priting	45.95	0.05	15.32	-	0.002	0.003	0.000	0.000
Beverage	-	-	-	-	46.3	79.7	2.6	1.0
Metal production	29,040.7	162,489.6	5773.6	44,943.9	104.04	167.29	0.04	0.02

**Table 5 ijerph-19-16156-t005:** Emission from households at different districts due to LPG and coal (tons/year).

No.	Districts	Households	Emission (Tons/Year)			
			Benzene	Toluene	Ethylbenzene	Xylene
1	District 1	48,408	0.37636	0.0695	0.027213	0.0107
2	District 2	36,792	0.000004	0.000002	0	0
3	District 3	49,083	0.0000081	0.000004	0	0
4	District 4	46,682	0.000007	0.000004	0	0
5	District 5	44,654	0.01365	0.0025	0.000987	0.0004
6	District 6	64,736	0.0000072	0.000004	0	0
7	District 7	77,545	0.08191	0.0151	0.005922	0.0023
8	District 8	107,992	0.000015	0.000008	0	0
9	District 9	72,655	1.26166	0.2329	0.091227	0.0359
10	District 10	59,640	0.08386	0.0155	0.006063	0.0024
11	District 11	57,649	0.0000077	0.000004	0	0
12	District 12	127,582	0.00976	0.0018	0.000705	0.0003
13	Go Vap	158,537	0.48232	0.089	0.034874	0.0137
14	Tan Binh	114,757	0.56681	0.1046	0.040984	0.0161
15	Tan Phu	116,123	0.000013	0.000007	0.00	0
16	Binh Thanh	121,996	1.5 × 10^−5^	0.000008	0	0
17	Phu Nhuan	45,619	0.0000061	3 × 10^−6^	0	0
18	Thu Duc	132,103	1.6 × 10^−5^	0.000008	0	0
19	Binh Tan	171,619	0.000022	1 × 10^−5^	0	0
20	Cu Chi	100,760	7.41391	1.3687	0.536082	0.211
21	Hoc Mon	105,618	0.000012	0.000006	0	0
22	Binh Chanh	147,863	5.94037	1.0967	0.429533	0.1691
23	Nha Be	34,806	0.0000032	0.000002	0	0
24	Can Gio	18,740	0.0000012	0.0000006	0	0
Total		2,061,959	16.2	3.0	1.2	0.5

**Table 6 ijerph-19-16156-t006:** Emission from restaurants in HCMC due to LPG and charcoal (tons/year).

	Emission from LPG (Tons/Year)				Emission from Charcoal (Tons/Year)			
Number of Restaurants	Benzene	Toluene	Ethylbenzene	Xylenes	Benzene	Toluene	Ethylbenzene	Xylenes
5069	0.000000442	0.000000223	-	-	3.7	0.7	0.3	0.1

**Table 7 ijerph-19-16156-t007:** Total BTEX emissions from the four main sources of air pollutants of BTEX in HCMC (tons/year).

Emission Sources	Benzene	Toluene	Ethylbenzene	Xylenes
Line	4387	7285	6321	5684
Non-road traffic (airport, railway)	2.56	1.11	0.03	2.78
Area	20	4	1	1
Point	29.4	163.1	5.8	45.0
Total (ton/year)	4435	7452	6328	5732

**Table 8 ijerph-19-16156-t008:** Contribution of some primary activities in BTEX emission in Ho Chi Minh City.

**Benzene**		**Toluene**	
Motorcycles	93%	Motorcycles	90%
Cars	5%	Cars	6%
LDVs	1%	LDVs	2%
Metal Production	1%	Metal Production	2%
**Ethylbenzene**		**Xylenes**	
Motorcycles	98.9%	Motorcycles	91.5%
Cars	0.8%	Cars	6.5%
LDVs	0.2%	LDVs	1.9%
Metal Production	0.1%	Metal Production	0.1%

## Data Availability

The datasets used during the current study are available.

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
