# Peer review of "An Innovative Method for BTEX Emission Inventory and Development of Mitigation Measures in Developing Countries—A Case Study: Ho Chi Minh City, Vietnam"

_ijerph, 2022, doi:10.3390/ijerph192316156_

Round 1
Reviewer 1 Report
The the emission of BTEX in a city was calculated and a method to calculate cold emission factor was develped in the paper. A methodology with bottom-up and top-down com- 20 bination was applied to conduct BTEX EI. The analysis results are interesting. There are two suggestions as follows,
1. Add the calculating time for Table 3. BTEX emission (kgs/year) from industrial sources due to producing and combustion activ- 242 ities in Ho Chi Minh city.
2. Add some testing materials to supprort the analysis result accuracy.
3. Expalin the reason why one group data is very different from other one in Table S1: Number of questionnaires during the campaign.
Author Response
Dear Associate editor and reviewer of IJERPH,
Thank you very much for your time and feedback. According to your comments, we have revised our manuscript titled “An innovative method for BTEX emission inventory and development of mitigation measures in developing countries” accordingly with carefully. The section below describes our responses (Bold sentences) based on each of your comment of Reviewer 1.
Moreover, we have edited other spelling mistakes for the whole manuscript. We look forward to receiving good news.
Sincerely,
On behalf of the authors,
Assoc. Prof. Ho Quoc Bang

Reviewer 2 Report
P.2, Line 85: Besides, the domestic cooking, other 84 transportation activities like airport, railways… also
It is better to change … for etc.
P.3: Numbers of the equations should be moved to the right.
The chapters are not correctly numbered.
P.4, Line 170: The EFs for Benzene, toluene….
Why is benzene with a capital letter?
P.5, Line 199: For emissions from area sources, the general formula is based on emission factors and activities…
Where is the formula or a reference to it?
Table 5: use dots or commas when separating digit order numbers.
P.10, Line 307: We indicated the contribution of primary pollution sources in Error! Reference 307 source not found…
P.11, Line 310: Cars contributed 5%, 6%, 0.8% and 6.5% of benzene, toluene, ethyl-benzene, and xylene, responsibility…
Respectively?
P.11, Line 321 (and further): Air smoke checking automobiles randomly on the road: HCMC has checked a few 321 days for buses: As a result…
Checkup punctuation!
P.12, Line 353: The results showed that transportation…
Maybe vehicle exhausts?
Author Response
Dear Associate editor and reviewer 2 of IJERPH,
Thank you very much for your time and feedback. According to your comments, we have revised our manuscript titled “An innovative method for BTEX emission inventory and development of mitigation measures in developing countries” accordingly with carefully. The section below describes our responses (Bold sentences) based on each of your comment of Reviewer 2.
Moreover, we have edited other spelling mistakes for the whole manuscript. We look forward to receiving good news.
Sincerely,
On behalf of the authors,
Assoc. Prof. Ho Quoc Bang
